# Plasma Circulating mRNA Profile for the Non-Invasive Diagnosis of Colorectal Cancer Using NanoString Technologies

**DOI:** 10.3390/ijms25053012

**Published:** 2024-03-05

**Authors:** Hin Fung Tsang, Xiao Meng Pei, Yin Kwan Evelyn Wong, Sze Chuen Cesar Wong

**Affiliations:** 1Department of Clinical Laboratory and Pathology, Hong Kong Adventist Hospital, Hong Kong SAR, China; andy_thf@yahoo.com.hk; 2Department of Health Technology and Informatics, The Hong Kong Polytechnic University, Hong Kong SAR, China; evelynwong0723@gmail.com; 3Department of Applied Biology & Chemical Technology, The Hong Kong Polytechnic University, Hong Kong SAR, China; xiaomeng2019.pei@polyu.edu.hk

**Keywords:** NanoString, nCounter, non-invasive screening, plasma RNA, CRC, colorectal cancer

## Abstract

Colorectal cancer (CRC) is one of the most prevalent cancers and the second leading cause of cancer deaths in developed countries. Early CRC may have no symptoms and symptoms usually appear with more advanced diseases. Regular screening can identify people who are at increased risk of CRC in order to offer earlier treatment. A cost-effective non-invasive platform for the screening and monitoring of CRC patients allows early detection and appropriate treatment of the disease, and the timely application of adjuvant therapy after surgical operation is needed. In this study, a cohort of 71 plasma samples that include 48 colonoscopy- and histopathology-confirmed CRC patients with TNM stages I to IV were recruited between 2017 and 2019. Plasma mRNA profiling was performed in CRC patients using NanoString nCounter. Normalized data were analyzed using a Mann–Whitney U test to determine statistically significant differences between samples from CRC patients and healthy subjects. A multiple-group comparison of clinical phenotypes was performed using the Kruskal–Wallis H test for statistically significant differences between multiple groups. Among the 27 selected circulating mRNA markers, all of them were found to be overexpressed (gene expression fold change > 2) in the plasma of patients from two or more CRC stages. In conclusion, NanoString-based targeted plasma CRC-associated mRNAs circulating the marker panel that can significantly distinguish CRC patients from a healthy population were developed for the non-invasive diagnosis of CRC using peripheral blood samples.

## 1. Introduction

CRC results from an abnormal growth of cells on the wall of the large bowel, which is the last portion of the digestive system. CRC is a complex disease, and it is believed to be formed by the accumulation of a cascade of genetic mutations in the body over time. Most CRC starts from polyps that are usually benign; however, some may develop into cancer over time after accumulating different mutations [1]. The development of a polyp into cancer may take more than 10 years. If it is not treated, cancer cells may spread to other parts of the body via the bloodstream and lymphatic system to invade and damage nearby organs. Adenoma–carcinoma sequence describes a gradual progression from normal epithelial mucosa to adenoma and then to carcinoma as a result of a series of genetic changes such as mutation and gene amplification [1]. Studies have shown that the risk of recurrence and subsequent death due to CRC is closely associated with the stage of the disease at the time of the first diagnosis [1,2]. Moreover, the risk of death from CRC could be reduced by detecting the disease at an earlier stage via mass screening and intervening [1,2].

With the growth of the aging population, it is expected that the number of cases, incidence, and death rates of CRC will increase further. Fortunately, CRC is highly curable if it is detected at an early stage. It is estimated that approximately 60% of all CRC deaths can be prevented by undergoing regular screening [3]. Early CRC may have no symptoms and symptoms usually appear with more advanced diseases. Common symptoms of CRC include a change in bowel habits (e.g., diarrhea or constipation) for unknown reasons that lasts for more than 2 weeks; persistent urge after passing stool; presence of blood or large amount of mucus in stool; abdominal discomfort (e.g., persistent pain, bloating, fullness, or cramps); weight loss and tiredness with unknown reason [1,2]. Direct visualization tests such as colonoscopy and sigmoidoscopy are currently the gold standards for examination of the lower digestive tract such as the colon and rectum. It allows clinicians to detect both cancerous and precancerous lesions by direct visualization. However, because colonoscopy involves invasive procedures, there is a risk of bowel perforation during colonoscopy and post-colonoscopy bleeding. Full bowel preparation and sedation are also required before the procedure. Moreover, the high examination cost as well as the risk of perforation during invasive procedures makes colonoscopy less suitable for a widespread population screening [4,5].

Human plasma contains RNA transcripts released by multiple cell types, including CRC cells. The potential application of serum circulating microRNA (miRNA) as biomarkers for CRC detection using the NanoString platform has been explored previously [6]. In this study, a NanoString-based targeted plasma circulating marker panel protocol for CRC-associated mRNAs in plasma was developed for the non-invasive diagnosis of CRC using peripheral blood samples. NanoString Technology utilizes novel digital color-coded barcode technology that is based on direct multiplexed measurement of gene expression and it offers high levels of precision and sensitivity. The use of our NanoString-based targeted plasma circulating marker panel allowed for the non-invasive detection of CRC, during which patients did not feel pain and embarrassment. Although our NanoString-based targeted plasma circulating marker panel cannot replace colonoscopy at this stage and colonoscopy is still the gold standard in the diagnosis of CRC, the results generated from this study will contribute to the development of a new targeted digital counting method without PCR amplification for early non-invasive CRC detection and monitoring of CRC patients.

## 2. Results

The CRC patients recruited in this study were aged between 50 and 88 (mean 71 ± 1) years old. The majority of the CRC patients were at stage III. Among the 27 selected candidate circulating mRNA markers, all of them were differentially expressed and able to distinguish CRC patients from healthy subjects significantly using plasma samples (*p* < 0.05). Those genes were found to be overexpressed (gene expression fold change >2) in the plasma of CRC patients in two or more CRC stages (Table 1). A heatmap showed a distinct gene expression pattern between CRC patients and healthy subjects without CRC (Figure 1). However, no significant difference in the expression of those genes from CRC TNM stages I to IV was found (Appendix A).

## 3. Discussion

Post-operative recurrence, metastasis, and spreading of tumor cells to lymph nodes are the major causes of cancer-related death in CRC patients. Prediction of the risk of tumor recurrence and metastasis is useful for guiding treatment decisions and disease management. Studies have shown that patients who receive early detection of disease recurrence and metastasis respond better to chemotherapy, and, thus, have better survival [7]. Hence, by identifying patients at high risk of recurrence and metastasis, the decision to adopt adjuvant treatment and close post-operative monitoring could be made immediately and appropriately in order to increase the survival rates of the patients. However, low sensitivity and low specificity for early-stage CRC detection using conventional screening methods such as the fecal occult blood test (FOBT) and fecal immunochemical test (FIT) [4,5], stool molecular tests [8,9], the blood-based carcinoembryonic antigen (CEA) [10] and *SEPT9* tests [11] are still a major concern.

In this study, we analyzed a cohort of 71 plasma samples that included 48 colonoscopy- and histopathology-confirmed CRC patients with TNM stages I to IV as well as 23 colonoscopy-confirmed healthy subjects as controls. We aim to develop a non-invasive test utilizing a NanoString-based platform to detect and monitor CRC. Patients can benefit from the early detection and appropriate treatment of the disease, as well as the timely application of adjuvant therapy after surgical operation. From the heatmap produced in this study, all 27 selected circulating mRNA markers were differentially expressed and able to distinguish CRC patients from healthy subjects significantly using plasma samples. Those genes were found to be overexpressed (gene expression fold change >2) in the plasma of CRC patients in two or more CRC stages. However, no significant difference in the expression of those genes between CRC TNM stages I and IV was found. One of the possible reasons for this could be that the small sample size limited us from evaluating the sensitivity and accuracy of the panel in each TNM stage accurately in this study. Therefore, further validation is required.

To the best of our knowledge, this study is the first study using NanoString technologies to measure the expression of a cohort of RNA biomarkers in the plasma of CRC patients. Our results showed that this system can detect significant differences between colonoscopy-confirmed healthy people and CRC patients. The selected circulating mRNA markers in this study include important markers that have been shown to be involved and over-expressed during CRC development [5,12,13,14,15,16,17,18,19,20,21,22,23,24,25,26,27]. Although potential CRC plasma circulating markers were identified in this study, this study has two limitations. First, the sample size was small. It prevented us from evaluating the sensitivity and accuracy of the panel in each TNM stage. An extensive evaluation with more CRC patients will be performed in the future. Second, the study was not designed to be age-matched between the CRC patients and healthy control subjects mainly because most CRC patients were diagnosed in the older age groups. An extensive evaluation with more age-matched patients and healthy subjects will be performed in the future to validate our results. In Hong Kong, the median age at diagnosis of CRC was 68 for males and 69 for females [28]. In order to focus on the over-expression of CRC-specific genes, we excluded patients who have been previously diagnosed and treated for CRC or the presence of synchronous malignancies. Therefore, it is not easy to find age-matched and colonoscopy-confirmed healthy control subjects for comparison in this study.

In conclusion, we have developed a NanoString-based targeted plasma circulating marker panel protocol for detecting CRC-associated mRNAs in plasma. This study is the first investigation to directly profile and analyze plasma mRNA markers in CRC patients utilizing a NanoString-based platform. The outcome of this study has also established a solid foundation for the non-invasive diagnosis of CRC using a peripheral blood sample. NanoString technology is a highly sensitive technology that can detect a scanty amount of RNA in plasma [29,30,31,32]. The results of this study have also shown that scanty amounts of RNA in plasma are detectable by NanoString technologies using random primers to amplify the RNA in plasma. This protocol can be applied in the development of non-invasive strategies or protocols for other cancers and diseases as well. Moreover, a panel of CRC-related markers was used for detection in this study. In order to translate this technology into routine clinical applications, a larger-scale study will be performed to include colorectal adenoma patients as well as patients with other common cancers such as breast cancer, liver cancer, and lung cancer to evaluate the specificity of this panel. Extensive evaluation of other factors such as logistics, regulation, and the cost of testing will also be needed to estimate the acceptance of this test in the future by the general public. Detection using multiple markers is more accurate than using individual single markers solely to predict the outcome of patients [33]. CRC detection using peripheral blood has long been an attractive approach because of its simplicity and non-invasive nature. The introduction of peripheral blood CRC screening has a potentially major impact on public health as well. For example, the Coronavirus Disease 2019 (COVID-19) pandemic has caused global changes in the delivery of healthcare services since 2019, including both outpatient community-based services and inpatient hospital-admission services [34]. During the COVID-19 pandemic, an accurate non-invasive CRC screening test could be an alternative to colonoscopy to reduce the risk of the transmission of infectious diseases and to relieve the workload in healthcare sectors [35,36].

However, in order to put this panel into clinical use, further clinical validation with a larger sample size is needed. Future work on this study can be focused on the follow-up on the plasma CRC-associated mRNA profiles in CRC patients (a) after surgical operation and before starting adjuvant therapy, (b) after completing adjuvant therapy, and through (c) tracing the change of plasma mRNA profiles for each patient to investigate if the observed changes are correlated with the patient’s clinical data such as metastasis, disease recurrence, and shorter survival to demonstrate the prognostic significance of the circulating marker panel.

## 4. Materials and Methods

### 4.1. Subjects Recruitment

In this study, a cohort of 71 plasma samples that include 48 colonoscopy- and histopathology-confirmed CRC patients with TNM stages from I to IV were recruited by the Departments of Surgery and Clinical Oncology, Queen Elizabeth Hospital between 1 September 2017 and 1 December 2019 with written informed consent. On the other hand, 23 colonoscopy-confirmed healthy subjects were recruited as healthy controls. The demographic and clinical staging information of the study population in this study are shown in Table 1. Ethics approval was obtained from the Kowloon Central Cluster Clinical Research Ethics Committee, Hong Kong (CREC Ref. No. 2019.542).

The inclusion criteria of this study were (1) patients who have been confirmed with CRC of various TNM stages, whereas the exclusion criteria of this study were (1) patients who have been previously diagnosed and treated for CRC, or (2) the presence of synchronous malignancies. Healthy volunteers with a family history of CRC, a history of colorectal polyps, cancer, inflammatory bowel, infectious diseases, or anemia were also excluded.

### 4.2. Customized CRC-Associated Targeted Plasma Circulating mRNA Markers Panel

Literature review was performed and 27 published circulating mRNA markers that have diagnostic and prognostic potential for CRC were selected. They include CEA, CK19, GCC, KIAA1199, TUG1, TRIM24, PIP4K2B, GK, BANK1, CDA, CTSL1, MMP9, Ki-67, c-MYC, FAP, BGN, INHBA, BCNP1, MS4A1, CTNNB, CK20, CDX2, S100A4, EPAS1, TYMS, MMP7, COX2, and 2 house-keeping genes (GAPDH and TBP) (Table 2). The target sequences for the design of CodeSets are shown in Appendix A. Figure 2 illustrates the framework of this study.

### 4.3. Blood Processing and Plasma RNA Extraction

For each subject, 9 mL of peripheral blood was collected in K3 EDTA tubes (Greiner Bio-one, Austria). Plasma was collected from peripheral blood samples using a double-centrifugation protocol described in our previous studies [45,46], which involves initial centrifugation at 1600× *g* followed by a second centrifugation at 16,000× *g*. Microfuge 22R centrifuge and an F301.5 rotor (Beckman Coulter) were used for centrifugation and plasma separation within 4 h after blood taking. After centrifugation, 4 mL of plasma was collected and preserved with 3.2 mL (0.8×) of Trizol (Life Technologies, Carsbad, CA, USA) before storage at −80 °C [45,46].

Cell-free RNA was extracted from plasma using our established protocol. In brief, 4.5 mL of preserved plasma with Trizol (Life Technologies, Carsbad, CA, USA) was used and the volume was topped up with Trizol to 5.5 mL. The plasma sample was then mixed with 2 mL of chloroform (Sigma–Aldrich, St. Louis, MO, USA), followed by centrifugation of 12,000× *g* for 15 min at 4 °C. The aqueous layer with RNA was collected and mixed with 1.5 volumes of absolute ethanol (Sigma–Aldrich, St. Louis, MO, USA) to achieve appropriate binding conditions. Total cell-free RNA was then extracted from the mixture using a miRNeasy Serum/Plasma kit (Qiagen, Hilden, Germany) according to the manufacturer’s protocol [47].

### 4.4. Targeted Genes Expression Profiling

The extracted RNA samples were then processed according to the manufacturer’s protocol for the nCounter low RNA input gene expression assay protocol.

### 4.5. cDNA Conversion

cDNA conversion was performed according to the manufacturer’s protocol of the nCounter Low RNA Input Amplification kit (NanoString Technologies, LOW-RNA-48, Seattle, WA, USA). In brief, the reverse transcription (RT) master mix was prepared by combining the RT Enzyme Mix and RT Primer Mix provided. For each sample, 1 μL of RT master mix was added to 4 μL of the extracted RNA sample. The cDNA conversion program was run on an Applied Biosystems 7500 Real-Time PCR System (Life Technologies) according to the manufacturer’s protocol. The concentrations of cDNA measured in this study are shown in Table 3.

### 4.6. Multiplexed Target Enrichment

The amplification master mix was prepared by combining 5X dT Amp Master Mix and gene-specific primers at 500 nM per primer. For each sample, 2.5 µL of PCR master mix was added to the converted cDNA sample. The multiplexed target enrichment program was run on an Applied Biosystems 7500 Real-Time PCR System (Life Technologies) for 8 amplification cycles according to the manufacturer’s protocol.

### 4.7. NanoString Digital Profiling

Sequence-specific oligonucleotide probes tagged with fluorescent barcodes were used to bind to and digitally measure cDNA. Hybridization was conducted for 21 h at 65 °C. Subsequently, probes were purified and immobilized on the nCounter Prep Station (NanoString Technologies). Each sample was scanned for 600 FOV (fields of view) on the nCounter Digital Analyzer (NanoString Technologies). Data were extracted using the nCounter RCC Collector. The abundance of specific capture probe-bound cDNA molecules was measured using the nCounter digital analyzer to count individual fluorescent barcodes. The NanoString pre-designed panel simultaneously detects 27 CRC-related genes, including 2 housekeeping transcripts. Six positive and eight negative spike-in controls, hybridization controls, and ligation-specific controls were included to determine sample integrity, quality, and background.

### 4.8. Data Analysis

All data analyses were performed using nSolver 4.0 (NanoString Technologies). The raw data from NanoString nCounter were normalized for lane-to-lane variation with a dilution series of 6 spike-in positive controls. The sum of the 6 positive controls for a given lane was divided by the average sum across lanes to yield a normalization factor, which was then multiplied by the raw counts in each lane to give normalized values. Codeset normalization was performed using the 2 housekeeping genes. For each sample, the mean plus 2 times the standard deviation (threshold = mean + 2SD) of the 8 negative controls (probes without target sequences) was subtracted from each miRNA count in that sample.

Normalization of data was achieved by eliminating digital counts below 50. Comparison of gene expression profiles between (a) CRC patients and healthy controls (Table 4) and (b) CRC patients of different stages was performed (Appendix A). A heatmap with statistically significant differences was generated. Hierarchical clustering was based on gene expression Z scores using the weighted pair group method with an averaging (WPGMA) algorithm (Figure 1). Multiple group comparison of clinical phenotype was performed using the Kruskal–Wallis H test for statistically significant differences between multiple groups and the Mann–Whitney U test for statistically significant differences between two groups. The p-values were adjusted using the Benjamini–Hochberg procedure. *p* ≤ 0.05 was taken as statistically significant.

## 5. Conclusions

A NanoString-based targeted plasma CRC-associated mRNAs circulating marker panel that can significantly distinguish CRC patients from healthy populations was developed for the non-invasive diagnosis of CRC using peripheral blood samples. It allows for early detection as well as the timely application of adjuvant therapy after surgical operation.

## Figures and Tables

**Figure 1 ijms-25-03012-f001:**
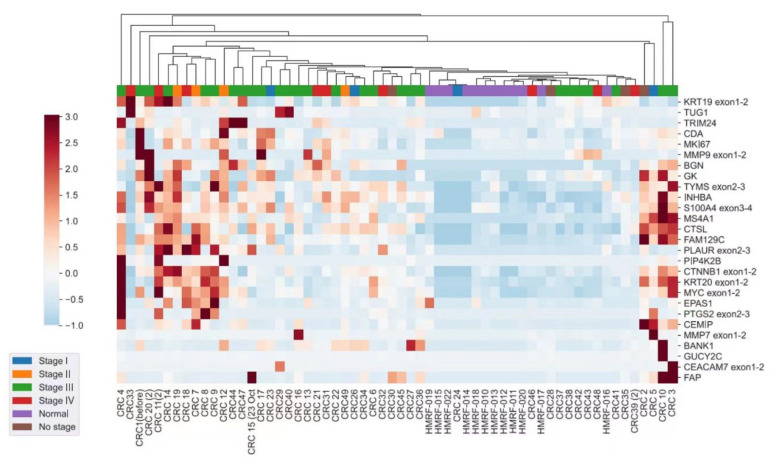
Heatmap of the expression profile in CRC patients of each TNM stage independent of that of the healthy controls. The vertical axis shows the gene names and the horizontal axis shows the various TNM stages of CRC patients and healthy subjects. Hierarchical clustering was based on gene expression Z scores using the WPGMA algorithm. The red and blue colors denote high and low intensities of gene expression, respectively.

**Figure 2 ijms-25-03012-f002:**
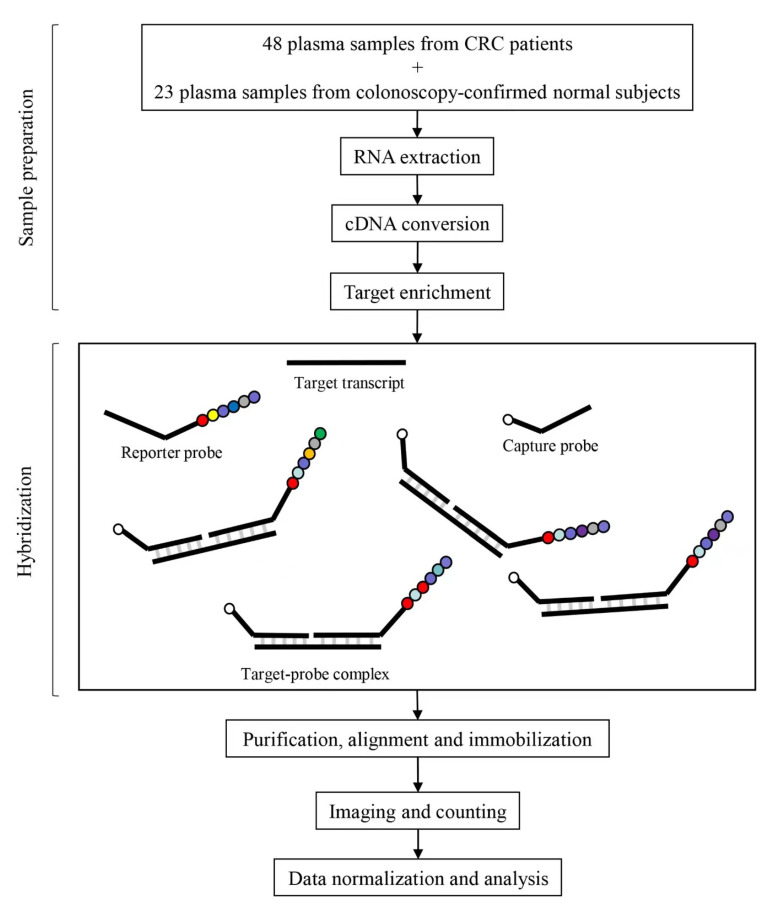
A schematic illustration of the framework using NanoString technology to detect CRC-related circulating mRNA markers in plasma samples. A customized CodeSet for the circulating mRNA markers’ panel consists of sequence-specific capture and reporter probes, each of 35–50 bases long. The CodeSets are hybridized to the target transcript. Following hybridization, the target-probes complexes are purified, aligned, and immobilized in the sample cartridge using the automated nCounter Prep Station. Cartridges are then transferred to the nCounter Digital Analyzer for imaging and direct digital counting of the molecular barcodes on the reporter probes.

**Table 1 ijms-25-03012-t001:** Demographic and clinical staging information of the study population.

Category	Patient (n)	Age (Mean ± SD)	Male (%)
Healthy	23	32.1 ± 10.7	48.7
CRC	48	71.1 ± 10.9	54.2
Stage I	4	69.5 ± 15.4	50.0
Stage II	4	64.5 ± 11.1	25.0
Stage III	27	71.2 ± 10.6	63.0
Stage IV	9	76.1 ± 9.1	55.6
No stage	4	66.8 ± 12.3	75.0

**Table 2 ijms-25-03012-t002:** Customized circulating mRNA markers panel composed of 27 published.

mRNA Markers	References
CEA, CK19, GCC	[12]
KIAA1199	[37]
TUG1, TRIM24, PIP4K2B, GK, BANK1, CDA, CTSL1, MMP9	[33]
Ki-67, c-MYC, FAP, BGN, INHBA	[38]
BCNP1, MS4A1	[39]
CTNNB, CK20, CDX2	[5,13,14]
S100A4	[40]
EPAS1	[41]
TYMS	[42]
MMP7, COX2	[43]
Housekeeping genes: GAPDH, TBP	[5,44]

**Table 3 ijms-25-03012-t003:** The concentrations of cDNA measured in this study.

Sample	Concentration (ng/µL)	Sample	Concentration (ng/µL)
1	2635.037	37	681.436
2	2463.077	38	647.449
3	2475.2	39	624.455
4	2285.758	40	639.05
5	2368.351	41	605.24
6	2435.414	42	608.247
7	2307.185	43	643.833
8	2217.029	44	734.516
9	2253.341	45	614.606
10	2322.118	46	609.581
11	2398.608	47	624.743
12	2334.615	48	611.096
13	2571.032	49	323.357
14	2392.264	50	306.63
15	2557.589	51	299.239
16	2388.509	52	306.919
17	2900.842	53	302.598
18	2738.253	54	331.867
19	2737.108	55	325.641
20	2884.761	56	305.816
21	2799.384	57	312.801
22	2783.456	58	304.714
23	2860.305	59	316.975
24	2749.807	60	331.7
25	2753.355	61	153.338
26	2535.082	62	156.764
27	2573.109	63	150.78
28	2591.199	64	171.028
29	2521.445	65	162.975
30	2483.813	66	162.26
31	2557.493	67	158.024
32	2799.152	68	161.375
33	2453.478	69	151.518
34	2474.785	70	150.675
35	2432.49	71	179.454
36	2392.586		

**Table 4 ijms-25-03012-t004:** Results of Kruskal–Wallis H test for significant differences between multiple groups. Twenty-seven differentially expressed RNAs were found to be upregulated in the plasma of CRC patients.

Gene Name	Refseq	Stage I	Stage II	Stage III	Stage IV	Normal	*p*-Value	Adjusted *p*-Value *
*BANK1*	NM_001083907.1	3118.818	3773.453	3697.386	944.5667	169.1133	0.001758	0.004394
*BGN*	NM_001711.3	103.0225	191.175	149.2763	123.5811	20	0.002165	0.004639
*CDA*	NM_001785.2	82,333	133,665.2	79,813.29	30,877.19	41,917.4	0.03538	0.04746
*CEACAM7*	NM_006890.4	46.1025	51.425	13,521.77	28.64333	20	0.010153	0.016922
*CEMIP*	NM_018689.1	47,200.59	59,372.89	23,834.92	18,151.22	4046.548	0.025741	0.035781
*CTNNB1*	NM_001098210.1	15,716.63	36,968.89	25,861.48	19,006.47	8143.892	0.003437	0.006751
*CTSL*	NM_001912.4	26,001.42	36,041.48	22,480.74	15,920.57	3366.493	0.003889	0.00713
*EPAS1*	NM_001430.3	283.21	3203.148	3505.034	1865.092	1235.516	0.024221	0.035057
*FAM129C*	NM_173544.4	58,697.87	99,605.14	59,054.28	47,203.52	16,909.75	0.000818	0.002248
*FAP*	NM_004460.2	321.02	102.3725	869.8074	46.50889	20	0.002193	0.004639
*GK*	NM_000167.3	19,534.83	29,865.46	20,696.59	10,550.17	3633.967	0.010934	0.017687
*GUCY2C*	NM_004963.1	61.775	97.025	494.8544	68.26778	20	0.002057	0.004639
*INHBA*	NM_002192.2	50,134.49	86,119.04	54,501	47,530.9	18,310.55	0.004839	0.008317
*KRT19*	NM_002276.4	12,402.46	32,193.63	26,178.84	29,384.17	4029.44	0.00057	0.001669
*KRT20*	NM_019010.2	46,344.5	11,6612.9	78,925.36	56,187.71	10,225.4	0.001336	0.003498
*MKI67*	NM_002417.2	56,987.24	74,004.38	77,026.36	52,881.03	22,054.26	0.026023	0.035781
*MMP7*	NM_002423.4	3649.235	277.8825	382.3	120.2722	20	0.00052	0.001669
*MMP9*	NM_004994.2	367.665	6548.393	7647.711	3684.182	20	0.000296	0.001086
*MS4A1*	NM_152866.2	18,589.93	22,468.3	17,110.81	7012.719	650.3958	0.000195	0.000893
*MYC*	NM_002467.4	13,616.51	34,941.22	27,283.9	26,276.21	5521.895	0.003054	0.006222
*PIP4K2B*	NM_003559.4	308.945	6175.323	1391.916	1687.758	20	0.002117	0.004639
*PLAUR*	NM_002659.3	565.765	12,562.74	6378.578	9516.01	1035.063	0.003746	0.007105
*PTGS2*	NM_000963.3	27,295.66	20,618.94	25,963.25	16,800.94	20	0.004645	0.008242
*S100A4*	NM_019554.2	33,689	44,015.95	37,234.61	25,325.74	18,203.82	0.012145	0.019084
*TRIM24*	NM_015905.2	592.65	8514.685	7197.833	3787.37	20	0.0000283	0.000398
*TUG1*	NR_002323.1	975.32	5631.128	8363.993	21,629.69	2689.192	0.014668	0.021804
*TYMS*	NM_001071.2	30,818.32	50,162.11	39,236.93	33,363.06	13,571.37	0.013251	0.020245

* adjusted *p*-value using the Benjamini–Hochberg procedure.

## Data Availability

The datasets used and/or analyzed during the current study are available from the corresponding author upon reasonable request.

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
