# Peer review of "Plasma Circulating mRNA Profile for the Non-Invasive Diagnosis of Colorectal Cancer Using NanoString Technologies"

_ijms, 2024, doi:10.3390/ijms25053012_

Round 1
Reviewer 1 Report
Comments and Suggestions for Authors
The study “Plasma circulating mRNA profile for the non-invasive diagnosis of colorectal cancer using NanoString technologies” by Hin Fung Tsang and team underscores the clinical relevance of this approach, emphasizing its potential for timely interventions and improved patient outcomes. Furthermore, the cost-effectiveness of the proposed diagnostic tool enhances its accessibility. However, the study falls short in data presentation, lacking specific quantitative results or figures. Additionally, the need for further validation studies and clinical trials, along with a more in-depth discussion on clinical integration and ethical considerations, represents notable areas for improvement. Overall, the NanoString-based approach shows promise, but addressing these concerns is vital for its successful translation into routine clinical practice.
1. The study acknowledges that the sample size is small, which may limit the generalizability of the findings. Small sample sizes can lead to statistical limitations and may not accurately represent the broader population.
2. The study mentions that due to the small sample size, the evaluation of sensitivity and accuracy of the identified panel in each TNM stage of CRC was limited. This raises concerns about the robustness of the findings across different stages of cancer.
3. The study was not designed to be age-matched between CRC patients and normal control subjects. This could introduce potential confounding factors, as age can impact gene expression and cancer risk. Age-matching would enhance the validity of the comparison between the two groups.
4. The study excluded patients who had been previously diagnosed and treated for CRC or had synchronous malignancies. This exclusion criteria might introduce selection bias and limit the generalizability of the findings to the broader population of CRC patients.
5. The study suggests potential applications for non-invasive CRC screening, especially during times like the COVID-19 pandemic. However, the translation of laboratory findings into routine clinical practice involves overcoming various logistical, regulatory, and cost-related challenges, which are not addressed in this study.
In conclusion, while the study contributes to the understanding of NanoString technology for CRC detection, the limitations highlighted above should be considered when interpreting the results.
Author Response
The study “Plasma circulating mRNA profile for the non-invasive diagnosis of colorectal cancer using NanoString technologies” by Hin Fung Tsang and team underscores the clinical relevance of this approach, emphasizing its potential for timely interventions and improved patient outcomes. Furthermore, the cost-effectiveness of the proposed diagnostic tool enhances its accessibility. However, the study falls short in data presentation, lacking specific quantitative results or figures. Additionally, the need for further validation studies and clinical trials, along with a more in-depth discussion on clinical integration and ethical considerations, represents notable areas for improvement. Overall, the NanoString-based approach shows promise, but addressing these concerns is vital for its successful translation into routine clinical practice.
- The study acknowledges that the sample size is small, which may limit the generalizability of the findings. Small sample sizes can lead to statistical limitations and may not accurately represent the broader population.
Response: Thank you for the comment. A total of 71 plasma samples that include 48 colonoscopy- and histopathology-confirmed CRC patients and 23 healthy people were recruited in this study. Multiple group comparison of clinical phenotype was performed by Kruskal-Wallis H test for statistically significant differences multiple groups and Mann-Whitney U test for statistically significant differences between two groups. The p-values were adjusted using Benjamini-Hochberg procedure and were also found to be statistically significant.
- The study mentions that due to the small sample size, the evaluation of sensitivity and accuracy of the identified panel in each TNM stage of CRC was limited. This raises concerns about the robustness of the findings across different stages of cancer.
Response: Thank you for the comment. Although we would like to proof that our panel can classify patients based on gene expression, our study can only distinguish CRC patients from non-CRC patients at this stage. This finding has been mentioned in the results section lines 90-93 : Heatmap showed a distinct gene expression pattern between CRC patients and normal subjects without CRC. (Figure 2) However, no significant difference in the expression of those genes from CRC TNM stages I to IV was found. (Supplementary Table 2)
- The study was not designed to be age-matched between CRC patients and normal control subjects. This could introduce potential confounding factors, as age can impact gene expression and cancer risk. Age-matching would enhance the validity of the comparison between the two groups.
Response: Thank you for the comment. We agree with the comment from the reviewer. It is because we would like to exclude patients who have been previously diagnosed and treated for CRC, or the presence of synchronous malignancies in order to focus on the over-expression of CRC-specific genes first as a proof-of-concept study. Therefore, it is not easy to find age-matched and colonoscopy-confirmed normal control subjects for comparison in this study. Further large-scale evaluation will be performed in the future.
- The study excluded patients who had been previously diagnosed and treated for CRC or had synchronous malignancies. This exclusion criteria might introduce selection bias and limit the generalizability of the findings to the broader population of CRC patients.
Response: Thank you for the comment. The exclusion criteria of this study were set to exclude patients who have been previously diagnosed and treated for CRC, or the presence of synchronous malignancies because we would like to focus on the over-expression of CRC-specific genes first as a proof-of-concept study. It has been explained in in lines 140-144. Further large-scale evaluation will be performed in the future.
- The study suggests potential applications for non-invasive CRC screening, especially during times like the COVID-19 pandemic. However, the translation of laboratory findings into routine clinical practice involves overcoming various logistical, regulatory, and cost-related challenges, which are not addressed in this study.
Response: Thank you for the comment. Related content has been added between lines 155-160: In order to translate this technology into routine clinical applications, a larger scale study will be performed to include colorectal adenoma patients as well as patients with other common cancers such as breast cancer, liver cancer and lung cancer to evaluate the specificity of this panel. Extensive evaluation on other factors such as logistics, regulation and the cost of testing will also be needed to estimate the acceptance of this test in future by the general public.
In conclusion, while the study contributes to the understanding of NanoString technology for CRC detection, the limitations highlighted above should be considered when interpreting the results.
Reviewer 2 Report
Comments and Suggestions for Authors
This article uses NanoString technology to explore the application of plasma circulating mRNA profiles in non-invasive diagnosis of colorectal cancer. The study analyzed 71 plasma samples and found that 27 circulating mRNA markers were overexpressed in the plasma of colorectal cancer patients. The research results indicate that the use of NanoString technology can develop a non-invasive biomarker combination for diagnosing colorectal cancer based on blood samples, which can significantly distinguish between colorectal cancer patients and healthy populations, and has certain significance. But there are some issues that need to be resolved.
1. This study used NanoString technology for non-invasive diagnosis of plasma circulating mRNA in colorectal cancer. The authors should elaborate on the advantages and limitations of this method compared to traditional diagnostic methods in the introduction section.
2. Researchers should conduct comparative gene expression profiling analysis to demonstrate that the manuscript's research can accurately distinguish between colorectal cancer patients and normal control groups.
3. Due to the small sample size, this study was unable to evaluate the sensitivity and accuracy of this panel in different TNM stages. The author's research needs to address this issue in order to better evaluate the application prospects of this panel in patients with different stages of colorectal cancer.
Author Response
This article uses NanoString technology to explore the application of plasma circulating mRNA profiles in non-invasive diagnosis of colorectal cancer. The study analyzed 71 plasma samples and found that 27 circulating mRNA markers were overexpressed in the plasma of colorectal cancer patients. The research results indicate that the use of NanoString technology can develop a non-invasive biomarker combination for diagnosing colorectal cancer based on blood samples, which can significantly distinguish between colorectal cancer patients and healthy populations, and has certain significance. But there are some issues that need to be resolved.
- This study used NanoString technology for non-invasive diagnosis of plasma circulating mRNA in colorectal cancer. The authors should elaborate on the advantages and limitations of this method compared to traditional diagnostic methods in the introduction section.
Response: Thank you for the comment. new content has been added between lines 76-80: The use of our NanoString-based targeted plasma circulating marker panel allows non-invasive detection of CRC, during which patients will not feel pain and embarrassed. Although our NanoString-based targeted plasma circulating marker panel cannot replace colonoscopy at this stage and colonoscopy is still the gold standard in the diagnosis of CRC, the results generated from this study will contribute to the development of a new targeted digital counting method without PCR amplification for early non-invasive CRC detection and monitoring of CRC patients.
- Researchers should conduct comparative gene expression profiling analysis to demonstrate that the manuscript's research can accurately distinguish between colorectal cancer patients and normal control groups.
Response: Thank you for the comment. Because of the large size of the data set, we put all the data in supplementary table 2.
- Due to the small sample size, this study was unable to evaluate the sensitivity and accuracy of this panel in different TNM stages. The author's research needs to address this issue in order to better evaluate the application prospects of this panel in patients with different stages of colorectal cancer.
Response: Thank you for the comment. Although we would like to proof that our panel can classify patients based on gene expression, our study can only distinguish CRC patients from non-CRC patients at this stage. This finding has been mentioned in the results section lines 90-93 : Heatmap showed a distinct gene expression pattern between CRC patients and normal subjects without CRC. (Figure 2) However, no significant difference in the expression of those genes from CRC TNM stages I to IV was found. (Supplementary Table 2)
Regarding the identification of CRC patients, multiple group comparison of clinical phenotype was performed by Kruskal-Wallis H test for statistically significant differences multiple groups and Mann-Whitney U test for statistically significant differences between two groups. The p-values were adjusted using Benjamini-Hochberg procedure and were also found to be statistically significant.
Reviewer 3 Report
Comments and Suggestions for Authors
The authors show that they can profile mRNA expression in plasma samples from patients with and without colorectal cancer as determined by colonoscopy using Nanostring Technology. There are significant differences in expression of a pre-selected group of genes with prognostic value for CRC between patients with and without confirmed CRC. There is no clear difference in expression of these genes between different Stages of CRC.
The study provides a nice proof of concept that mRNA profiling from plasma using Nanostring technologies is possible and may prove to be useful in non-invasively detecting CRC. However, the conclusions that are drawn are not completely supported by the data.
In the abstract it is stated that the authors have developed a "NanoString-based targeted plasma CRC-associated mRNAs circulating marker panel that can significantly distinguish CRC patients from healthy population". However, the tests that the authors perform only show that the selected genes are, on average, significantly differentially expressed between healthy subjects and CRC patients. Figure 2 shows that the authors are not able to completely distinguish between CRC and healthy, because a lot of CRC patients cluster with and some even in the middle of, the healthy controls.
If the authors want to support their claims they should suggest a method to classify patients based on their gene expression profile and describe how well it performs. This should be possible with the data they describe here. In general, the results section is very short. Various analyses that are described in the methods are not shown in the results, which is a puzzling.
The description of all the genes in the discussion is not very helpful. If they were shown to be relevant in the results section, further description in the discussion would be warranted.
Comments on the Quality of English LanguageNo comments.
Author Response
The authors show that they can profile mRNA expression in plasma samples from patients with and without colorectal cancer as determined by colonoscopy using Nanostring Technology. There are significant differences in expression of a pre-selected group of genes with prognostic value for CRC between patients with and without confirmed CRC. There is no clear difference in expression of these genes between different Stages of CRC.
Response: Thank you for the comment. Although we would like to proof that our panel can classify patients based on gene expression, our study can only distinguish CRC patients from non-CRC patients at this stage. Further large-scale evaluation will be performed in the future.
The study provides a nice proof of concept that mRNA profiling from plasma using Nanostring technologies is possible and may prove to be useful in non-invasively detecting CRC. However, the conclusions that are drawn are not completely supported by the data.
In the abstract it is stated that the authors have developed a "NanoString-based targeted plasma CRC-associated mRNAs circulating marker panel that can significantly distinguish CRC patients from healthy population". However, the tests that the authors perform only show that the selected genes are, on average, significantly differentially expressed between healthy subjects and CRC patients. Figure 2 shows that the authors are not able to completely distinguish between CRC and healthy, because a lot of CRC patients cluster with and some even in the middle of, the healthy controls.
Response: Thank you for the comment. We agree with the comment from the reviewer. Although we would like to proof that our panel can classify patients based on gene expression, our study can only distinguish CRC patients from non-CRC patients at this stage. This finding has been mentioned in the results section lines 90-93 : Heatmap showed a distinct gene expression pattern between CRC patients and normal subjects without CRC. (Figure 2) However, no significant difference in the expression of those genes from CRC TNM stages I to IV was found. (Supplementary Table 2)
If the authors want to support their claims they should suggest a method to classify patients based on their gene expression profile and describe how well it performs. This should be possible with the data they describe here. In general, the results section is very short. Various analyses that are described in the methods are not shown in the results, which is a puzzling.
Response: Thank you for the comment. Because of the large size of the data set, we put all the data in supplementary table 2. Although we would like to proof that our panel can classify patients based on gene expression, our study can only distinguish CRC patients from non-CRC patients at this stage. This finding has been mentioned in the results section lines 90-93 : Heatmap showed a distinct gene expression pattern between CRC patients and normal subjects without CRC. (Figure 2) However, no significant difference in the expression of those genes from CRC TNM stages I to IV was found. (Supplementary Table 2)
The description of all the genes in the discussion is not very helpful. If they were shown to be relevant in the results section, further description in the discussion would be warranted.
Response: Thank you for the comment. The description of the genes in discussion has been removed.
Round 2
Reviewer 1 Report
Comments and Suggestions for Authors
I am satisfied with the responses.
Reviewer 2 Report
Comments and Suggestions for Authors
The manuscript has been sufficiently improved to warrant publication in IJMS.